# Reliable New Biomarkers of Mitochondrial Oxidative Stress and Neuroinflammation in Cerebrospinal Fluid and Plasma from Alzheimer’s Disease Patients: A Pilot Study

**DOI:** 10.3390/ijms26167792

**Published:** 2025-08-12

**Authors:** Rosa Di Lorenzo, Chiara Zecca, Guglielmina Chimienti, Tiziana Latronico, Grazia Maria Liuzzi, Vito Pesce, Maria Teresa Dell’Abate, Francesco Borlizzi, Alessia Giugno, Daniele Urso, Giancarlo Logroscino, Angela Maria Serena Lezza

**Affiliations:** 1Department of Biosciences Biotechnologies and Environment, University of Bari Aldo Moro, Via Orabona 4, 70125 Bari, Italy; rosa.dilorenzo@uniba.it (R.D.L.); guglielminaalessandra.chimienti@uniba.it (G.C.); tiziana.latronico@uniba.it (T.L.); angelamariaserena.lezza@uniba.it (A.M.S.L.); 2Center for Neurodegenerative Diseases and the Aging Brain, Department of Clinical Research in Neurology, University of Bari Aldo Moro, “Pia Fondazione Cardinale G. Panico”, 73039 Tricase, Italy; czecca@piafondazionepanico.it (C.Z.); dellabatemariateresa@gmail.com (M.T.D.); francescoborlizzi91@gmail.com (F.B.); giugnoalessia89@gmail.com (A.G.); d.urso@piafondazionepanico.it (D.U.); giancarlo.logroscino@uniba.it (G.L.); 3Department of Translational Biomedicine and Neuroscience (DiBraiN), University of Bari Aldo Moro, 70124 Bari, Italy

**Keywords:** mitochondrial oxidative stress, neuroinflammation, Alzheimer’s disease, cerebrospinal fluid and plasma biomarkers, superoxide dismutase 2, cell free mitochondrial DNA, DNase I, matrix metalloproteinases, sex-stratification analysis

## Abstract

Mitochondrial oxidative stress and neuroinflammation are involved in the onset and progression of Alzheimer’s disease (AD). Novel reliable, circulating biomarkers related to these processes were searched in cerebrospinal fluid (CSF) and plasma samples. Paired CSF and plasma samples from 20 subjective memory complaints (SMC) subjects, 20 mild cognitive impairment (MCI) due to AD subjects, and 20 Alzheimer’s dementia (ADd) patients were analyzed. Protein amounts of manganese-containing superoxide dismutase 2 (SOD2), cell-free mitochondrial DNA (cf-mtDNA) level, DNase I, and matrix metalloproteinases 2 and 9 (MMP-2 and MMP-9) activities were determined. As for SOD2, an MCI male-specific significant increase in both biofluids and an ADd male-specific significant decrease in plasma were found. No significant differences were demonstrated in cf-mtDNA level. An ADd-specific significant increase in plasma DNase I and MMP-2 activities was found. A SMC female-specific significant higher value in CSF MMP-9 activity in comparison to male counterparts was demonstrated. The present results suggest a male patient-specific (MCI and ADd) regulation of SOD2 expression in plasma and support an ADd-specific increase in plasma DNase I and MMP-2 activities. Therefore, the potential of SOD2 amount, DNase I, and MMP-2 activities in plasma as new markers of ADd should be explored. The SMC female-specific high activity of MMP-9 might contribute to AD female-sex bias.

## 1. Introduction

Alzheimer’s disease (AD), the most common cause of dementia, is a progressive age-related neurological disorder involving multiple pathological processes, including neuronal loss, synaptic dysfunction, neuroinflammation, the accumulation of amyloid plaques and neurofibrillary tangles, alterations in lipid and bioenergetics metabolism, oxidative stress, and vascular dysregulation [1]. Although the “amyloid cascade hypothesis” well addresses several pathogenic features [2,3], it no longer appears to be enough to clarify AD onset and progression. Thus, huge attention has been paid by researchers to understanding the etiology and pathogenesis of this multifactorial disease with several risk factors [4]. The great contribution of oxidative stress to the initiation and progression of AD has been acknowledged [5]. An imbalance between oxidants, mainly reactive oxygen species (ROS) and reactive nitrogen species (RNS); physiological products of cellular metabolism; and antioxidants lead to a shift from the beneficial role of the reactive species as molecules involved in cell signaling [6] and protection against infectious organisms [7] to harmful mediators of cellular damage [7]. Aging and AD are both marked by defects in brain metabolism and increased oxidative stress, albeit to varying degrees and, indeed, aging represents the main “risk” factor of AD [8]. Mitochondria are involved in both physiological aging [9,10] and AD [11] by controlling cellular bioenergetics and redox homeostasis. The age-related decrease in mitochondrial energy production and increase in oxidative stress should eventually exceed a threshold after the failure of compensatory mechanisms. The central role of mitochondria in the degenerative mechanisms is the basis of the “Alzheimer mitochondrial cascade hypothesis” [12], which postulates that mitochondrial function may affect amyloid precursor protein (APP) expression and processing as well as β Amyloid (Aβ) accumulation. Such hypothesis could explain also the pathogenesis of AD cases not fitting with the “amyloid cascade hypothesis” [2,3]. Effectively, mitochondrial deficits have been already detected before the presence of Aβ deposits [13], and cognitive impairments might begin many years, even decades, before the open onset of the disease, subtly setting the conditions for the overt manifestation. It is now accepted that the neuropathological cascade progresses along a continuum of disease from a long preclinical phase, characterized by normal cognition and abnormal brain biomarkers, to mild cognitive impairment (MCI) and then to clinically apparent AD dementia (ADd) [14,15,16]. The initial neurocognitive symptoms of preclinical AD have a neurobiological basis and can be monitored through the assessment of biological molecules reflecting pathological changes in human fluids decades before disease’s onset [17,18]. It is noteworthy to remember that AD manifestations also extend outside the brain, and the systemic abnormalities might be not only secondary to cerebral events but reflect underpinning processes in the course of the disease [19]. In agreement with the “Alzheimer mitochondrial cascade hypothesis” [11], mitochondrial oxidative stress has been reported in different kinds of samples from AD patients, as well as animal and cell models [20], and it might be related to the well-characterized age-related mitochondrial oxidative stress [21,22]. The latter can be counteracted by antioxidants present inside the organelles, such as the manganese-containing superoxide dismutase 2 (SOD2) enzyme, which represents one of the major cellular antioxidant defenses systems [23,24]. According to the role of mitochondrial dysfunction in AD [11,20], the expression of SOD2, which is involved in maintaining mitochondrial homeostasis [25], has been a subject of investigation. The relevance of the increased mRNA amounts of SOD2 found in lymphocytes [26] and of the reduced protein contents evidenced in both neurons and astrocytes [27] of AD patients supports the important role of oxidative damage and impaired antioxidant response in the disease. Moreover, the significance of changes in SOD2 amounts, as a suitable biomarker in the context of the mitochondrial signature in AD, has been further confirmed by the ultra-deep proteomic analysis of brain tissue and biofluids [28,29]. Mitochondria expand their functions well beyond their role as cell energy plants, being involved in intra- and extracellular signaling and communication [30]. In particular, dysfunctional mitochondria might release mitochondrial DNA (mtDNA) into the cytoplasm or in the extracellular space as cell-free mtDNA (cf-mtDNA). Indeed, the content of cf-mtDNA may act as a marker of mitochondrial oxidative stress and of a related pathological condition, as demonstrated in different kinds of diseases [31,32,33], and might constitute a stress signal in AD [34]. Released mtDNA could function as a highly immunogenic damage-associated molecular pattern (DAMP), acting as a mediator of “sterile inflammation” [35]. Decreased contents of cf-mtDNA in cerebrospinal fluid (CSF) have been considered an early biomarker for AD [36], possibly secondary to the decreased contents of brain cellular mtDNA typical of neurodegenerative diseases [37]. On the other hand, increased plasma cf-mtDNA has been associated with aging and AD, in spite of decreased blood cellular mtDNA copy numbers [34]. Circulating cf-DNA is cleared by DNases, mainly DNase I, waste-management enzymes secreted into body fluids [38]. Interest on the counteracting role of this enzyme against the oxidatively damaged mtDNA, a possible driver of Aβ peptide aggregation [39] and neuroinflammation/degeneration [40], has risen [41].

The immunogenic activity of mtDAMPs, released by dysfunctional mitochondria and recognized by microglial immune receptors, contributes to neuroinflammation progression, thus demonstrating the crosstalk between mitochondrial dysfunction and neuroinflammation [42]. The continuous activation of microglia interacts with mitochondrial dysfunction and, through the promotion of neurodegeneration, leads to neuronal death [43]. Matrix metalloproteinases (MMPs), a family of zinc (Zn^2+^)-containing endopeptidases, are expressed in the brain by neurons, astrocytes, and microglia. Their activity and expression inside cells are tightly regulated by different factors such as ROS and pro-inflammatory cytokines as well as by tissue inhibitors of MMPs (TIMPs). MMPs are involved in various biological and pathological processes in the central nervous system (CNS). As for their role in the progression of neurodegeneration, MMPs are involved in neuroinflammation, microglial activation, blood brain barrier (BBB) disruption, dopaminergic apoptosis, and α-synuclein modulation [44]. In particular, MMP-2 and MMP-9 are able to trigger several neuroinflammatory and neurodegenerative pathways and have been involved in AD progression. In fact, MMP-2 and MMP-9 have been suggested to contribute to BBB damage through their release from BBB-proximal cells such as microglia, pericytes, and infiltrating leukocytes. Together with MMP-9, MMP-2 appears to promote and fine-tune neuroinflammatory processes such as the expression of chemokines and pro-inflammatory cytokines, and large increases in MMP-2 over time have been associated with enhanced brain atrophy [45]. Furthermore, increased MMP-2 and MMP-9 activities as well as expression have been reported in specific brain areas, depending on the different stages of AD. However, results in the literature are not consistent; thus, the determination of the exact role of any specific MMP in the AD pathophysiology is still to be pursued, but it holds promise for establishing their role as potential biomarkers reflecting the severity or progression of the disease [46]. On the other hand, MMPs are involved in many physiological processes in the CNS, such as cell migration, the regulation of neurite outgrowth, and neuronal plasticity. Among the beneficial functions of MMPs, their ability to degrade Aβ is very relevant. MMP-2 and MMP-9 are able to cleave Aβ monomers and oligomers, while MMP-9 is unique in its ability to also cleave Aβ fibrils and clear plaques from amyloid-laden brains, thus making MMPs potential targets for neurodegenerative diseases and in particular AD [47].

The search for reliable circulating biomarkers of mitochondrial oxidative stress and neuroinflammation should include both CSF and plasma since their identification in plasma would make it possible to reduce invasive procedures such as lumbar puncture. In consideration of this relevant purpose, in the present study, paired CSF and plasma samples from subjects with diagnosis of subjective memory complaints (SMC), patients with mild cognitive impairment (MCI) due to AD and patients with AD dementia (ADd) have been analyzed. The major aim of this work was to further characterize mitochondrial involvement and neuroinflammation development in the AD progression through the identification of novel mitochondrial and neuroinflammatory biomarkers, which might be added to the routinely assessed ones. The amount of SOD2 protein, identified as a marker of mitochondrial oxidative stress, was compared in CSF and plasma. Furthermore, the number of copies/μL of cf-mtDNA was quantified in both biofluids, and the activity of DNase I was evaluated in the CSF/plasma of the same subjects. In addition, the activity of MMP-2 and of MMP-9 has also been assessed to analyze markers related to the development of neuroinflammation.

Furthermore, analyses about a sex-related influence on the markers have also been carried out to obtain additional pieces of information about the verified female-sex bias of AD. In fact, two-thirds of AD patients are women, and the sudden drop in estrogen levels after the menopause has been proposed to be one risk factor in AD [48].

## 2. Results

### 2.1. Demographic and Clinical Data

The study included 60 participants from SMC (*n* = 20), MCI (*n* = 20) and ADd (*n* = 20) groups. Demographic and clinical data of the participants are shown in Table 1.

The age range was 39–81 years for SMC, 50–88 years for MCI, and 36–79 years for ADd. Sex composition and age were not statistically different between the three groups. CSF amyloid, t-tau, and p-tau levels were highly significantly different between groups and remained so also when values were binned according to sex in each group (Table 1).

### 2.2. SOD2 Amount in CSF and Plasma Samples

SOD2 protein amounts were analyzed in paired samples of CSF and plasma from the three groups of subjects through western immunoblotting.

In CSF, SOD2 amounts were not significantly different between the three groups, with median values being increased by 65% and 40% in MCI and ADd, respectively, in comparison with SMC (Figure 1, panel A). In plasma, a statistically significant (*p*: 0.0044, Kruskal–Wallis test) decrease was shown in ADd in comparison with SMC (60% decrease) and with MCI (18% decrease). The Dunn’s multiple test revealed statistically significant differences between ADd and both the SMC (*p* < 0.01) and MCI groups (*p*: 0.05) (Figure 1, panel B).

When SOD2 values were analyzed according to sex, a significant difference was found in CSF samples in males (*p*: 0.0473, Kruskall–Wallis test) but not in females. The Dunn’s post test showed a statistically significant increase in male MCI with respects to SMC (*p* < 0.05) (Figure 1, panel C). Also, in plasma samples, the difference was statistically significant in the male group (*p*: 0.0014, Kruskall–Wallis test) but not in the female group. The Dunn’s post test showed a statistically significant decrease in ADd patients compared to the SMC (*p* < 0.05) and to MCI (*p* < 0.01) groups (Figure 1, panel D). To further investigate the possible influence of sex, the Mann–Whitney test was applied to compare data from male and female subjects belonging to the same study groups. The test was quite statistically significant concerning the CSF SOD2 values in the MCI group (*p*: 0.0630, Mann–Whitney test), and statistically significant (*p*: 0.0147, Mann–Whitney test) when plasma values were compared in the same MCI group (Figure 1, panels E and F, respectively).

The Spearman r correlation test was applied to search for possible correlations between SOD2 amounts and the assessed biomarkers of AD (Ab, t-tau, and p-tau) in CSF samples. A statistically significant positive correlation was found only with t-tau (r = 0.6224, *p* = 0.0347) and p-tau (r = 0.6294, *p* = 0.0323) proteins, respectively, in male ADd patients (Figure 2).

### 2.3. Cf-mtDNA in CSF and Plasma Samples

Cf-mtDNA was evaluated in paired samples of CSF and plasma from the enrolled subjects by dd-PCR. Median cf-mtDNA copies/mL were higher in plasma than in CSF samples in all study groups, being 12-, 9-, and 19-fold higher, respectively, in the SMC, MCI, and ADd groups (Figure 3, panels A and B).

Cf-mtDNA copies/μL were not significantly different between the three groups neither in CSF nor in plasma samples. In CSF, the median cf-mtDNA copy number was 24% increased and 16% reduced, respectively, in MCI and ADd patients, in comparison with SMC (Figure 3, panel A). Plasma samples from MCI patients showed a 6% decreased median value, and those from ADd patients showed a 29% increased value compared with that from SMC subjects (Figure 3, panel B). When values were binned and analyzed according to sex, no statistically significant difference emerged between the three groups of subjects (Figure 3, panels C and D). When data from male and female subjects belonging to the same group were compared, no statistical significance emerged (Figure 3, panels E and F).

The Spearman r correlation test was applied to search for possible correlations between the cf-mtDNA copy number and assessed biomarkers of AD (Ab, t-tau, and p-tau) in CSF samples. A close-to-statistically significant positive correlation was found only with t-tau (r = 0.7143, *p* = 0.0576) protein in female ADd patients (Figure 4).

### 2.4. DNase Activity in CSF and Plasma Samples

DNase activity was evaluated in paired samples of CSF and plasma from the enrolled subjects through a single radial enzyme diffusion assay (Figure 5, panels A and B).

Median DNase activity/mL in CSF showed a 12% decrease in MCI patients and a 53% increase in ADd, in comparison with the SMC group. The difference was close to statistical significance (*p*: 0.0536, Kruskal–Wallis test) (Figure 5, panel A). Median DNase activity/mL in plasma was increased by 0.3% in MCI and increased by 18% in ADd patients, in comparison with the SMC group. The difference was statistically significant (*p*: 0.0123, Kruskal–Wallis test). The Dunn’s multiple post test revealed a statistically significant marked increase in ADd in comparison with SMC, which was also present in the comparison between the ADd and MCI groups (*p*: 0.0270, SMC vs. ADd; *p*: 0.0350, MCI vs. ADd) (Figure 5, panel B). When values were binned and analyzed according to sex, a statistically significant difference was present in plasma samples from female ADd patients (*p*: 0.0340, Kruskal–Wallis test). The Dunn’s post test revealed a statistically significant increase in the ADd vs SMC groups (*p* < 0.05) (Figure 5, panel D). The ADd condition featured a circa doubled median value of DNase activity in comparison with the MCI counterpart in CSF.

When data from male and female subjects belonging to the same group were compared, no statistical significance emerged (Figure 5, panels E and F).

The Spearman r correlation test was applied to search for possible correlations between DNase activity and assessed biomarkers of AD (Ab, t-tau, and p-tau) in CSF. A statistically significant negative correlation was found only with t-tau (r = −0.7426, *p* = 0.0074) and p-tau (r = −0.6200, *p* = 0.0350) proteins, respectively, in ADd male patients (Figure 6).

### 2.5. MMP-2 and MMP-9 Activities in CSF and Plasma Samples

The MMP-2 and MMP-9 activities in samples from the subjects under investigation were evaluated by zymographic analysis.

As shown in Figure 7, panel A, MMP-2 activity was not significantly different between the three groups in CSF, whereas there was a statistically significant difference between the plasma samples (*p*: 0.0472, one-way ANOVA). The Dunnett’s post-test evidenced a statistically significant increase (17%) in ADd patients in comparison with SMC subjects (*p* < 0.05) (Figure 7, panel B). When values were binned and analyzed according to sex, no statistically significant difference emerged (Figure 7, panels C and D), although the median CSF value in male MCI patients appeared to be 21% reduced compared with that in SMC subjects (Figure 7, panel C). When data from male and female subjects belonging to the same group were compared, no statistical significance emerged, neither in CSF nor in plasma. However, in CSF samples from MCI, the median value in males was 16% lower than that in females, whereas this difference was nuanced (8%) in plasma (Figure 7, panels E and F). The MMP-2 plasma values in ADd were the highest for this biofluid both in males and females.

Although MMP-9 activity was not significantly different between the three study groups in CSF, a 5% and 40% increase was detected in MCI and ADd patients, respectively, compared to SMC (Figure 8, panel A). Also, in plasma samples, a non-significant increase in MCI and ADd patients appeared (80% and 40%, respectively) (Figure 8, panel B). When CSF samples were analyzed according to sex, median values appeared higher in male subjects both in the MCI and ADd groups than in SMC (78% and 66% increase, respectively), whereas in the female MCI patient group, the median was 50% lower than that in the SMC and Add groups (Figure 8, panel C). In plasma samples, the behavior in males was comparable to that in CSF, showing an increase in the MCI (133%) and ADd groups (76%), compared to SMC. A different behavior was observed in plasma samples from females, since after a slight increase in the MCI group, MMP-9 in the ADd group decreased to an activity lower than that detected in the SMC group (Figure 8, panel D). When data from male and female subjects belonging to the same group were compared, the difference was statistically significant in the CSF of the SMC group (*p*: 0.0359, *t*-test) (Figure 8, panel E). This sex influence was evident also in plasma samples, although it was not statistically significant (Figure 8, panel F).

In CSF samples from MCI, median value in males was 56% higher than in females, whereas this difference was nuanced (8%) in plasma (Figure 8, panels E and F). It is remarkable that the MMP-9 values from female SMC subjects were higher than the male counterparts in both CSF and plasma, suggesting a sex-related influence in the basal level of this protease.

The Pearson r correlation test was applied to search for possible correlations between MMP activity and the assessed biomarkers of AD (Ab, t-tau, and p-tau) in CSF. A statistically significant negative correlation was found only between MMP-9 and Ab values in SMC and MCI male subjects in CSF samples (r = −0.6895, *p* = 0.0399, in SMC subjects; r = −0.6378, *p* = 0.0473, in MCI patients.) (Figure 9).

## 3. Discussion

The major aim of the present study was the in-depth further characterization of mitochondrial involvement in AD onset and development, as well as of the neuroinflammation role in AD progression. Such aim was pursued through the search of reliable, circulating biomarkers of mitochondrial oxidative stress and neuroinflammation, which might be added to the routinely assessed AD laboratory markers. Another goal of this study, sought through the comparison between the results in CSF and plasma samples, was the identification of plasma markers for diagnostical/therapeutical purposes, reducing invasive CSF sampling. Therefore, paired CSF and plasma samples from 20 subjects with subjective memory complaints (SMC), 20 patients with mild cognitive impairment (MCI) due to AD, and 20 patients with AD dementia (ADd) have been analyzed. The size of the studied groups was quite small and, due to the very large interindividual variability in humans, the analyses in the present study often did not reach statistical significance, but our priority was to complete this pilot study, meant to pave the way for further works including larger sizes and, eventually, also longitudinally analyzed groups of sampled subjects. Furthermore, analyses about a sex-related influence on the markers have also been carried out to shed some more light on the verified female-sex bias of AD. The SOD2 protein was chosen as an early and reliable marker of mitochondrial oxidative stress [49,50], likely affected by AD [28,29]. Different trends of SOD2 amounts in the analyzed biofluids were found. In fact, in CSF, there was no disease-related significant change in the expressed protein among the three compared groups of SMC subjects, MCI patients, and ADd patients, not being stratified according to sex. However, when SOD2 values were stratified, a significant increase was found in male MCI samples with respect to the SMC counterparts. Male ADd patients showed a trend towards an increase in the protein amount in comparison with SMC subjects. The absence of changes among the female groups could suggest a failure to counteract oxidative stress, in line with the higher prevalence of the pathology in females than in males. Indeed, this suggestion could be sustained by a study performed in an animal model, reporting data about sex-related differences in *SOD2* gene expression in distinct regions of the mouse central nervous system. The authors proposed that male mice were more prone to activate endogenous antioxidant systems at the mitochondrial level [51]. Effectively, a recent study also highlighted a sex-component in the pathogenesis of AD, shown by a decrease in the levels of the neuroprotective factor irisin in the CSF of female AD patients, and suggested that it might be linked to sex hormone fluctuations after menopause [52]. A sex-related influence on SOD2 expression was further indicated in the present study by the statistically significant positive correlation found with t-tau and p-tau proteins in ADd male patients. The relevance of the mitochondrial antioxidant response through the regulation of SOD2 expression has also been clearly demonstrated in a very recent study including controls, AD patients, and “nondemented with Alzheimer’s neuropathology” (NDAN) individuals. The analysis of postmortem frontal cortices revealed a dramatic oxidative damage, associated with a significant decrease of total SOD2 levels in the AD brain. By contrast, NDAN individuals showed low oxidative damage and an efficient antioxidant response, possibly cooperating with their cognitive intactness. In particular, SOD2 was selectively induced in NDAN neurons [27]. Results from the present work show that in plasma, there was a disease-related significant change in the SOD2 protein amount among the three compared groups. Notably, all group values were circa one order of magnitude higher than the CSF counterparts, with a statistically significant progressive decrease from SMC subjects through MCI patients to ADd patients. The increased SOD2 amount found in male MCI plasma was mirroring the CSF counterpart and was suggestive of an attempt to induce an antioxidant compensatory response, supporting a sex-related regulation of SOD2 in CSF and plasma. The relevant significant SOD2 decrease in male ADd plasma was consistent with the similar trend in CSF and was highly ADd-specific, suggesting that the SOD2 amount in plasma might confidently be considered a new reliable biomarker to be added to the set of those routinely tested for diagnostical purposes. This finding indicated that the mitochondrial oxidative stress, involved in AD onset and progression, was present at an early stage of the disease also in districts different from the CNS and might constitute an underpinning process leading to the systemic abnormalities of the disease. As for the analysis in female plasma, the decrease in SOD2 amount was highly specific of the disease condition since it was shared by MCI and ADd patients and it might be very likely induced by some sex-related, maybe hormonal, mechanisms. Mitochondrial dysfunction and the consequent oxidative stress have been reported in various kinds of AD samples, as well as animal and cell models [11], and it has been suggested that increased levels of ROS might drive mtDNA release from mitochondria into cytoplasm and from the cell into the extracellular space and circulation as cell-free molecules (cf-mtDNA) or encapsulated in extracellular vesicles (EVs) [34]. EVs include mitochondrial-derived vesicles (MDVs), which shuttle mitochondrial constituents to other organelles or cells [53], protecting their load from degrading activities. The number of copies of cf-mtDNA, determined in CSF, and eventually in plasma, has been proposed as a potential biomarker of AD onset and/or progression. The determination of cf-mtDNA in CSF has led to contrasting results and such discrepancies; in spite of the same experimental technique used, namely dd-PCR [36,54,55], cf-mtDNA in CSF has been related to various causes [37,54,55]. In the present study, the number of copies/μL of cf-mtDNA was determined by dd-PCR in both biofluids, and generally, the values were one order of magnitude higher in plasma than in CSF samples in all study groups. No significant difference among the three groups was found in CSF or in plasma samples, with all groups featuring very dispersed values, as already reported for CSF by [55]. Individual values of cf-mtDNA copies/μL largely overlapped between SMC subjects and MCI or ADd patients, thus preventing the actual use of this parameter as a diagnostic biomarker. As for the possible correlations between cf-mtDNA copy number in CSF and the assessed biomarkers of AD (Aβ, t-tau, and p-tau), a close-to-statistically significant positive correlation was found only with the t-tau protein in ADd female patients. Such results might suggest a very heavy mitochondrial oxidative stress in these patients, leading to a more severe neuronal loss, characterized by an increased release of cf-mtDNA and rise of t-tau, previously indicated as a marker of neuronal damage [36] in CSF. This conclusion might be supported by the absence of compensatory mitochondrial responses, namely by the absence of an increase in SOD2 expression in female MCI and ADd patients. MtDNA and other molecules extruded from mitochondria may act as a stress signal related to mitochondrial dysfunction. Clearance of extracellular DNA is performed by DNases, mostly by DNase I, which hydrolyzes DNA in the blood and prevents the induction of inflammation. To our knowledge, only one study has detected DNAse I activity in CSF from an animal model of pneumococcal meningitis [56] and the evaluation of the enzyme activity as a potential marker in patients biofluids is a novel finding of the present study. A preliminary connection between post-traumatic increased concentration of extracellular mtDNA, decreased DNase I activity, and the development of systemic inflammatory response syndrome (SIRS) has been hypothesized in patients who underwent major surgical interventions, leading to the suggestion of DNase administration to modulate the severity of SIRS [57]. In the present study, DNase I activity/mL was assessed in the CSF and plasma from all groups. The comparison among groups in CSF showed a close-to-statistically significant 53% increase in ADd, which was evenly present in both sexes, thus suggesting no sex-related influence. The assessment of plasma values demonstrated a statistically significant increase in DNase I activity in ADd patients, mirroring the CSF behavior. When results were analyzed according to sex, the ADd-specific significant increase was found in female ADd patients, but a similar trend was also evident in male ADd patients. Therefore, the determination of DNase I activity in plasma from ADd patients might be suggested as a new reliable, disease-specific biomarker, useful for diagnostical purposes. A statistically significant negative correlation was found with t-tau (r = −0.7426, *p* = 0.0074) and p-tau (r = −0.6200, *p* = 0.0350) proteins, respectively, in ADd male patients. Interestingly, aiming to develop a therapy for neurodegeneration, in a very recent study performed in a mouse model of age-related neurodegeneration, EVs loaded with recombinant DNAse I were delivered to brain microglia in vivo. The EV-mediated elimination of cytosolic dsDNAs was sufficient to prevent neuroinflammation, reduce neuronal apoptosis, and delay the onset of neurodegenerative symptoms in the model mice [40], thus suggesting a new therapeutical approach also for AD neurodegeneration. Effectively, the unique, somewhat empirical therapeutical use of DNase I in an end-stage AD patient was also previously reported, leading to a rapid, considerable improvement in cognitive and behavioral functions [58].

Neuroinflammation has been largely demonstrated to be very relevant for AD progression, and MMPs have been suggested to be important in AD. In particular, MMPs have both detrimental and beneficial functions in neuroinflammation and neurodegeneration since they contribute to microglia activation and BBB disruption, although on the other hand, several MMPs are able to degrade Aβ aggregates and might thus promote the cerebral clearance of Aβ, slowing AD progression. The relatively low concentrations of Aβ in brain extracellular fluids (interstitial fluid, ISF, and CSF) in normal physiological conditions are due to a balance between Aβ biosynthesis and clearance. Failure of clearance mechanisms rather than overproduction of Aβ appears to be involved in the development of late-onset AD. Since MMP-2 and MMP-9 are able to degrade Aβ, their regulation has been indicated as a therapeutic target in AD drug development [59]. Furthermore, animal models deficient in MMP-2 and/or MMP-9 showed significant increases in brain steady-state levels of Aβ, leading to the suggestion that MMP-2 and MMP-9, secreted from astrocytes, may contribute to extracellular Aβ clearance [60]. However, an increase in MMP-2 and MMP-9 expressions might be problematic because of possible damage to the BBB [61]. In the present study, MMP-2 and MMP-9 activities have been determined in CSF and plasma samples, allowing to us to obtain novel results which can be added to those of the few studies determining the MMPs activities/levels in both the plasma and CSF of AD patients [62,63]. The results of this study showed a sort of linear progression of MMP-2 plasma activity in agreement with the severity of the disease condition that led to a statistically significant increase of MMP-2 in ADd patients in comparison with SMC subjects. Such increase was suggestive of a disease-specific change that might be verified in plasma for diagnostical/therapeutical purposes. Different results depending on the assessed biofluid were also reported for MMP-9. MMP-9 CSF activity revealed a marked tendency of the enzyme to increase in ADd patients, although without reaching statistical significance, whereas the plasma MMP-9 activity in MCI showed a peculiar, marked increase that exceeded both values from the SMC and ADd groups. Furthermore, the enzyme activity was higher in the CSF and plasma from ADd patients in comparison to the counterpart values from SMC patients, supporting the ADd-specific increase of MMP-9 in plasma, as already reported by other groups [64,65,66]. Interesting considerations can be made by sex-stratification analysis of MMP-9 activity, since there was a novel result showing a statistically significant higher level of this enzyme in CSF from female SMC patients in comparison with male counterparts. This novel finding supports a differential, sex-related activity of MMP-9 in CNS, which is mirrored, although without statistical significance, in plasma. Previous reports suggested that sex might modulate MMPs activity in AD patients. Post-menopausal women may be particularly vulnerable to AD-related pathological changes because of the absence of the neuro- and vaso-protective effects of estrogen, relevant to the AD progression and including MMP-9 pathways [67]. Furthermore, estrogens modulate neuronal MMP-2 and MMP-9, which can degrade Aβ in cell cultures [68]. The above-quoted study by Tsiknia et al. [67] demonstrated that higher MMP-9 plasma concentrations predicted a faster worsening of cognitive functions only in women. The same study also suggested that MMP-9 might be involved in potential mechanisms contributing to women’s elevated susceptibility to AD. Consistent with such suggestion, the significantly higher level in MMP-9 activity, observed in this study in the CSF from SMC females in comparison with SMC males, might contribute to predispose women to a faster onset and development of the disease. Indeed, in males, a marked increase in MMP-9 was observed in more advanced stages of the disease. A positive cross-interaction between Aβ, MMP-2, and MMP-9 has been suggested through results from in vitro and ex vivo studies, as well as steady-state mouse models, and Aβ has also been shown to induce MMP-9 expression and activity in vitro in astrocytes [69]. However, the inverse correlation between MMP-9 and Aβ, observed in the CSF from SMC and MCI males, is suggestive of a sex-specific effect of MMP-9 in the first stages of the disease, possibly related to the ability of MMP-9 to degrade Aβ. When the disease condition was overt, as in ADd patients of both sexes, this effect was not evident. Although there is no exact consensus about MMPs values measured in the blood or CSF of SMC, MCI, and ADd subjects, the results of this study add useful information to further clarify the role of MMPs in AD, especially with regard to the sex-specific differences observed in the CSF. Therefore, their presence in a multimarker panel could be justified to predict the development and progression of AD [46]. The same is true for the mitochondrial oxidative stress markers of SOD2 amounts and DNase activity, which should be also present in a multimarker panel helpful for early diagnosis and for less invasive sampling in the follow-up of diagnosed patients. Furthermore, the determination of the above-discussed markers in plasma might cooperate in the diagnosis of those cases not fitting in the usual presentation of AD-related CSF biomarkers.

## 4. Materials and Methods

### 4.1. Study Design

This is a cross-sectional study according to the Strengthening the Reporting of Observational Studies in Epidemiology (STROBE) reporting guideline [70], including 60 subjects outpatients who consecutively referred to the Center for Neurodegenerative Diseases and the Aging Brain of the University of Study of Bari “Aldo Moro” at Pia Fondazione “Card. Panico” Hospital (Tricase), over a period from January 2016 to December 2021: 20 subjects, 11 female and 9 male, with SMC (subjective memory complaints); 20 patients, 10 female and 10 male, affected by MCI (mild cognitive impairment) due to AD; and 20 patients, 8 female and 12 male, affected by ADd (Alzheimer’s disease dementia), were enrolled in this study.

All patients underwent a multidisciplinary assessment with neurological and neuropsychological examinations, nutritional assessment, 3T magnetic resonance imaging (MRI) scan, routine laboratory assessment and lumbar puncture for cerebrospinal fluid (CSF) analysis, and/or 18F-florbetaben (FBB) positron emission tomography (PET) as part of the diagnostic procedure. Demographic data, including age, sex, and years of education, were collected. A structured interview exploring familiar, personal and medical history, and social status, as well as an overall physical exam, were performed.

A neurological examination was carried out, including a standard neurological exam and the Clinical Dementia Rating Scale (CDR), a tool designed to grade subjects from normal function through various stages of dementia, assessing six cognitive and functional domains, including memory, orientation, judgment, community affairs, home hobbies, and personal care. The Mini Mental State Examination (MMSE) was used to exclude impairment of global cognition.

The diagnosis of AD dementia was performed using the Diagnostic and Statistical Manual of Mental Disorders, 5th ed. (DSM-5) [71], and National Institute on Aging-Alzheimer’s Association (NIA-AA) criteria [72].

Patients with MCI were selected according to the following criteria: (1) cognitive concern reflecting a change in cognition reported by patient or informant or clinician; (2) impairment in one or more of four cognitive domains from the neuropsychological test battery; (3) normal functional activities as derived from the CDR and the Functional Activities Questionnaire; (4) the absence of dementia (DSM-IV).

Patients diagnosed with SMC presented subjective memory concern and were “self-referrals”. These patients were cognitively normal with no significant impairment in cognitive functions or activities of daily living. Criteria for diagnosis were: (1) self-experienced persistent decline in memory and cognitive capacity in comparison with a previously normal status and unrelated to an acute event; (2) normal age-, gender-, and education-adjusted performance on standardized cognitive tests, which are used to classify mild cognitive impairment (MCI) or prodromal AD [73].

Exclusion criteria included comorbid neurological or any major psychiatric disease; drug abuse; clinical or neuroimaging evidence of focal lesions; and/or inflammatory, infectious, or vascular diseases. Subjects who showed poor adherence to study procedures and had a poor or incomplete medical history were also excluded.

All study participants gave their written informed consent, and the study was approved by the Local Ethical Committee (ASL Lecce verbale n°6, 25 May 2017), according to the Declaration of Helsinki 1991 [74]. Subjects were selected based solely on clinical diagnosis without discrimination by age, sex, or race.

### 4.2. Sample Collection and CSF AD Biomarkers Analysis

All patients underwent lumbar puncture according to standard procedures. The CSF sample was centrifuged at room temperature for 10 min at 2000× *g*, aliquoted, and stored at −80 °C until analysis, according to international biomarkers recommendations [75].

Venous blood was drawn by venipuncture from all patients; blood samples were collected in EDTA vacutainers, which were immediately centrifuged for 15 min at ~2000× *g* at room temperature within 1 h. After centrifugation, plasma was removed, aliquoted (0.5 mL/aliquot) into screw-cap polypropylene tubes, and stored at −80 °C until biochemical analyses.

The CSF Aβ42, t-tau, and p-tau181 levels were measured by chemiluminescent immunoassay CLEIA (Lumipulse G ß- amyloid 1–42, Lumipulse G Total Tau, Lumipulse G pTau181, Fujirebio Europe N.V., Gent, Belgium) on a fully automatic platform (Lumipulse G600II, Fujirebio Europe N.V., Gent, Belgium). All the assays were performed according to the manufacturer’s protocols. For the interpretation of the CSF biomarker results, the following cut-off values were considered: Aβ42 > 599 pg/mL, t-tau < 3 42 pg/mL, p-tau181 < 57 pg/mL, p-tau181/Aβ42 < 0.08.

### 4.3. Cf-mtDNA Quantification

Cf-mtDNA levels (copies/μL) in paired samples of CSF and plasma were quantified using digital droplet PCR (ddPCR). Used hydrolysis probe and primers, complementary to *CYTB* mtDNA gene, were according to [76]. PCR assay conditions were as in [77]. The QX200 ddPCR system (Bio-Rad Laboratories Inc., Hercules, CA, USA) was employed. Briefly, 2 μL of CSF and 2 μL of 10-fold diluted plasma were used in a total reaction volume of 22 μL containing 11 μL of 2× ddPCR Supermix for probes (without dUTPs), 0.9 μM of each primer, and 0.1 μM of the FAM-labeled probe. A minimum of 10,000 droplets were generated by the means of the Bio-Rad Automated Droplet Generator (Bio-Rad Laboratories Inc., Hercules, CA, USA) and subjected to PCR as follows: initial denaturation at 95 °C for 10 min, 40 cycles of denaturation at 95 °C for 15 s and annealing/extension at 60 °C for 1 min (ramp rate: 2.5 °C per second), and final extension at 98 °C for 10 min. The QX200 reader was used to analyze PCR-subjected droplets, and data were interpreted using the QX Manager Software 2.0 (Bio-Rad Laboratories Inc., Hercules, CA, USA).

### 4.4. Measurement of DNase Activity

A single radial enzyme diffusion assay [78] was used to evaluate total DNase activity in paired CSF (25 μL) and plasma (5 μL) samples. The procedure was as in [77]. In brief, calf thymus DNA (Sigma-Aldrich, St. Louis, MO, USA) was added at 100 μg/mL in assay buffer (35 mM Tris-HCl pH 7.8, 20 mM MgCl_2_, 2 mM CaCl_2_) containing 2.5× GelRed (Sigma-Aldrich, St. Louis, MO, USA) as fluorescent DNA dye. After 10 min heating (50 °C), the solution was mixed with a same volume of 2% agarose in water and allowed to solidify in plastic tray.

Samples or dilutions of recombinant DNase I (Promega Corporation, Madison, WI, USA), of known concentrations, were loaded into the gel. After overnight incubation (37 °C), the diameters of circles on gel, representing the remaining fluorescence of hydrolyzed DNA, were measured using the ChemiDoc System and Image Lab 6.1 software (Bio-Rad Laboratories Inc., Hercules, CA, USA).

### 4.5. Gel Electrophoresis and SOD2 Western Blotting

Proteins contained in CSF and plasma were separated by sodium dodecyl sulphate–polyacrylamide gel electrophoresis (SDS-PAGE). First, 20 μL of CSF and 2 μL of plasma were loaded on 4–15% Criterion^TM^ TGX Stain-Free^TM^ Precast Gels (#5678085, Bio-Rad Laboratories, Inc., Hercules, CA, USA). Protein bands were transferred on polyvinylidene fluoride (PVDF) membranes. The blots were blocked with 5% bovine serum albumin (BSA), 1× Tris-buffered saline, 0.1% Tween-20 and then incubated with SOD2 primary antibody (MnSOD, Cell Signaling Technology Inc., Danvers, MA, USA #13194S; dilution 1:250) overnight at 4 °C.

Immunoreactive bands were detected with secondary goat anti-rabbit conjugated with horseradish peroxidase (Anti-rabbit IgG, HRP-linked Antibody, Cell Signaling Technology Inc., Danvers, MA, USA #7074; dilution 1:5000) incubated for 1 h and then developed using the Clarity^TM^ Western ECL Substrate (#170-5061, Bio-Rad Laboratories, Inc., Hercules, CA, USA) with a ChemiDoc System (Bio-Rad Laboratories, Inc., Hercules, CA, USA). The obtained immunofluorescent bands were normalized to the total protein content in each lane using the stain-free technology gels [79]. The densitometric analysis was carried out using the Bio-Rad Image Lab Software^TM^ 6.0.1 (Bio-Rad Laboratories, Inc., Hercules, CA, USA).

### 4.6. Detection of MMP Activity in Plasma and CSF Samples

Plasma and CSF activities of MMP-2 and MMP-9 were determined by SDS-PAGE zymography as already described [80,81]. Briefly, 40 μL of undiluted CSF samples were precipitated in cold acetone for 1 h at −20 °C, then centrifuged, and the pellets were dissolved in 10 μL of loading buffer containing SDS. One microliter of plasma samples was directly dissolved in 10 μL of electrophoresis loading buffer. The solubilized samples were separated on 10% polyacrylamide gels copolymerized with 0.1% (*w*/*v*) gelatin (Sigma Chemical Co., St. Louis, MO, USA). After electrophoresis, the gels were washed in a buffer containing 2.5% (*w*/*v*) Triton X-100 and then incubated for 24 h at 37 °C in 1% (*w*/*v*) Triton X-100 in the presence of 10 mM CaCl_2_, pH 7.4. After staining, the gels were analyzed by scanning densitometry using Bio-Rad Image Lab Software^TM^ 6.0.1 (Bio-Rad Laboratories, Inc., Hercules, CA, USA), and MMP-2 and MMP-9 activities were expressed as optical density (OD) × mm^2^.

### 4.7. Statistics

Subjects’ characteristics are described by mean ± SD, whereas clinical data are medians with 25–75th interquartile range. All data represent the results of at least two independent experiments and are shown as individual values with medians. The Kolmogorov–Smirnov test was used to evaluate normal or non-normal distribution. Differences in sex composition were assessed using the χ^2^ test. Data were analyzed by the *t*-test, Mann–Whitney test, or one-way ANOVA with Tukey multiple comparison test or the Kruskal–Wallis Test with Dunn’s multiple comparison test for normal or non-normal distributed variables, respectively. Spearman’s correlation test was used for the correlation analysis. Statistics was performed with Prism statistical software release 8.0.1 (GraphPad Software, San Diego, CA, USA).

## 5. Conclusions

This study confirmed the relevant role of mitochondrial oxidative stress and neuroinflammation in AD onset and progression. Novel, interesting results were obtained through the analysis of some enzymes and molecules involved in these processes, carried out in CSF and plasma from SMC subjects, MCI subjects, and ADd patients. The small size of the studied groups likely prevented the verification of the condition of the plasma mirroring the CSF counterpart in the comparison between the behaviors of the suggested potential markers. Nevertheless, the present results open scientific pathways and debates that may be confirmed or not through future studies requiring larger sizes of sample groups, in order to assess the reliability of the suggested markers for diagnostical/therapeutical purposes and to reduce invasive CSF sampling. The first limitation of the present study is inherent to its being a pilot, single-center, open-label study with a clear explorative purpose, providing only preliminary evidence. Further studies with a greater cohort of patients as well as of healthy controls are needed to support these initial observations and claims with statistical significance. Second, this is a cross-sectional study; therefore, the biomarkers were only detected once, and further, large-scale longitudinal studies need to be conducted to demonstrate the dynamics of SOD2, DNase, MMP-2, and MMP-9 alterations along the spectrum of AD and their correlation with clinical and biological markers. Furthermore, in the present study, analyses about a sex-related influence on the markers have also been carried out to shed some more light on the female-sex bias of AD. In particular, a sex-related influence was proposed for the mitochondrial compensatory increased expression of SOD2 found in male MCI patients, which was abolished in ADd male patients and absent in female patients. The negative correlation found between MMP-9 and Ab values in CSF from SMC and MCI male subjects was suggestive of a male sex-specific beneficial effect of MMP-9 in the first stages of the disease.

## Figures and Tables

**Figure 1 ijms-26-07792-f001:**
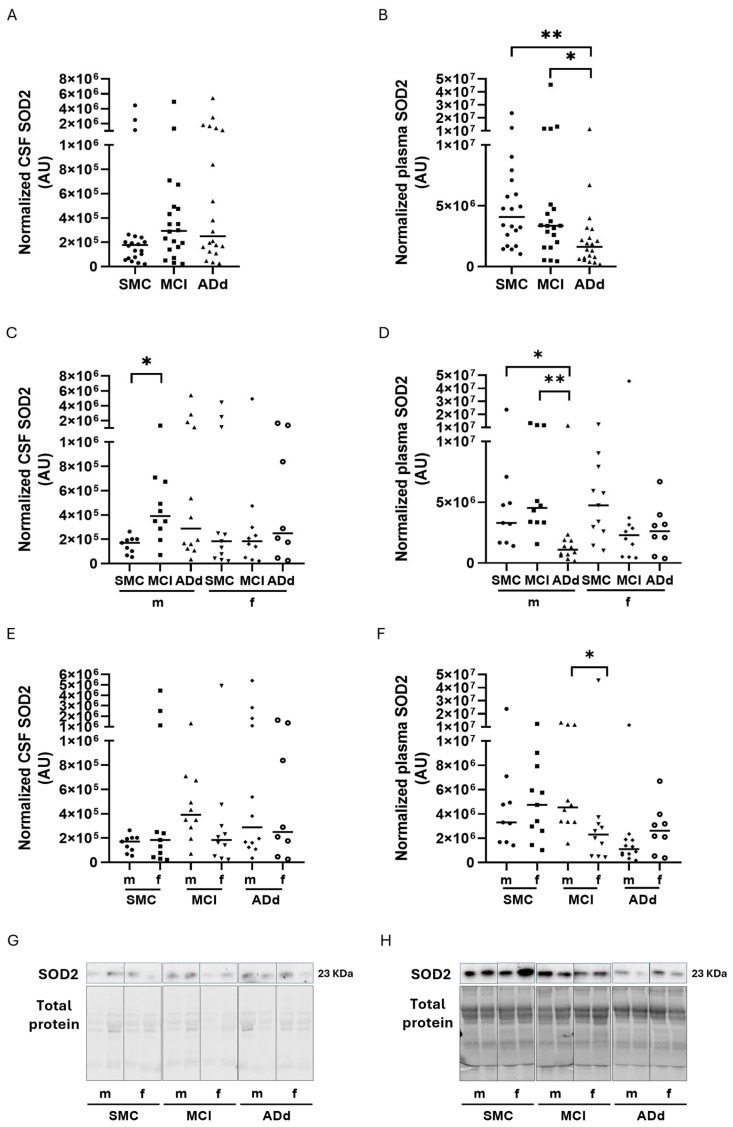
Normalized SOD2 values in CSF and plasma samples in the three study groups. Data are individual values with medians. Panels (**A**,**B**): SOD2 values in CSF and plasma, respectively, from 20 SMC subjects, 20 MCI subjects, and 20 ADd patients. Panels (**C**,**D**): SOD2 values in CSF and plasma, respectively, according to sex in the three groups. Panels (**E**,**F**): SOD2 values in CSF and plasma, respectively, in subjects dichotomized according to sex in each study group. Panels (**G**,**H**): representative blots of Stain-Free membranes, loaded with CSF (**G**) and plasma (**H**) samples (see Appendix A). Panels (**A**–**D**): * = *p* < 0.05, ** = *p* < 0.01, Dunn’s post test. Panel (**F**): * = *p* < 0.05, Mann–Whitney test. SMC = subjective memory complaints. MCI = mild cognitive impairment. ADd = Alzheimer dementia. CSF = cerebrospinal fluid. m = male group: SMC, *n* = 9; MCI, *n* = 10; ADd, *n* = 12. f = female group: SMC, *n* = 11; MCI, *n* = 10; ADd, *n* = 8.

**Figure 2 ijms-26-07792-f002:**
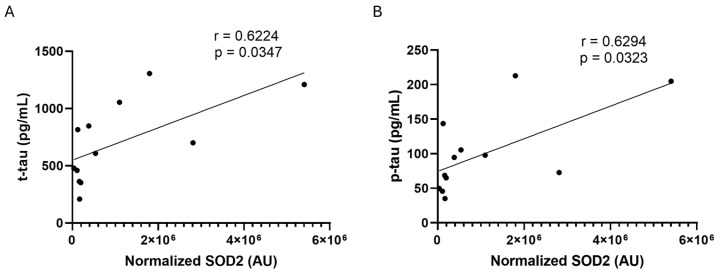
Regression analysis. Spearman’s correlation test between SOD2 values and t-tau (panel (**A**)) and p-tau (panel (**B**)) in male ADd patients in CSF samples. *n* = 12.

**Figure 3 ijms-26-07792-f003:**
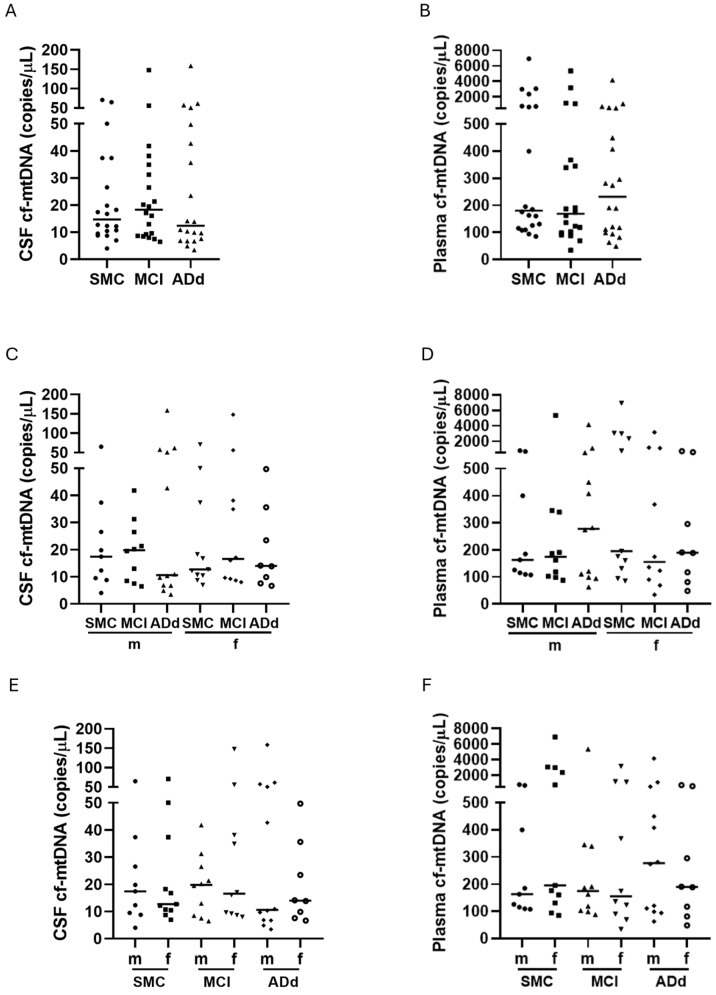
Levels of cf-mtDNA in CSF and plasma samples in the three study groups. Data are individual values with medians. Panels (**A**,**B**): cf-mtDNA copies/mL in CSF and plasma, respectively, from 20 SMC subjects, 20 MCI subjects, and 20 ADd patients. Panels (**C**,**D**): cf-mtDNA copies/mL in CSF and plasma, respectively, according to sex in the three groups. Panels (**E**,**F**): cf-mtDNA in CSF and plasma, respectively, in subjects dichotomized according to sex in each study group. SMC = subjective memory complaints. MCI = mild cognitive impairment. ADd = Alzheimer dementia. CSF = cerebrospinal fluid. m = male group: SMC, *n* = 9; MCI, *n* = 10; ADd, *n* = 12. f = female group: SMC, *n* = 11; MCI, *n* = 10; ADd, *n* = 8.

**Figure 4 ijms-26-07792-f004:**
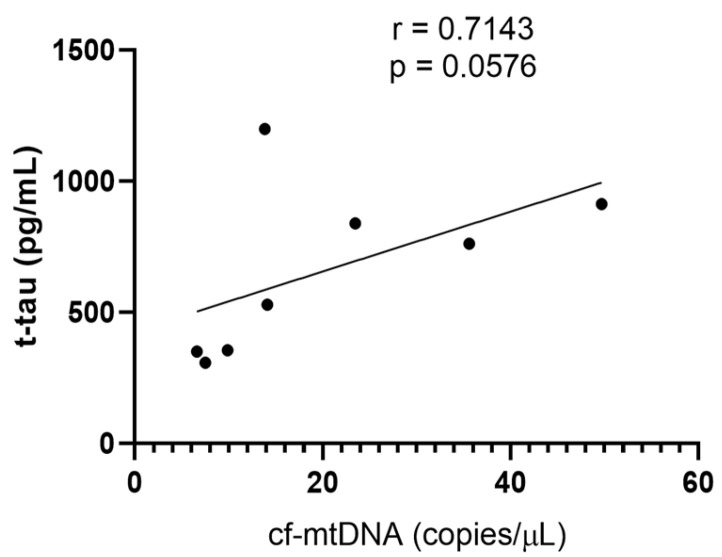
Regression analysis. Spearman’s correlation test between cf-mtDNA and t-tau in female ADd patients in CSF samples. *n* = 8.

**Figure 5 ijms-26-07792-f005:**
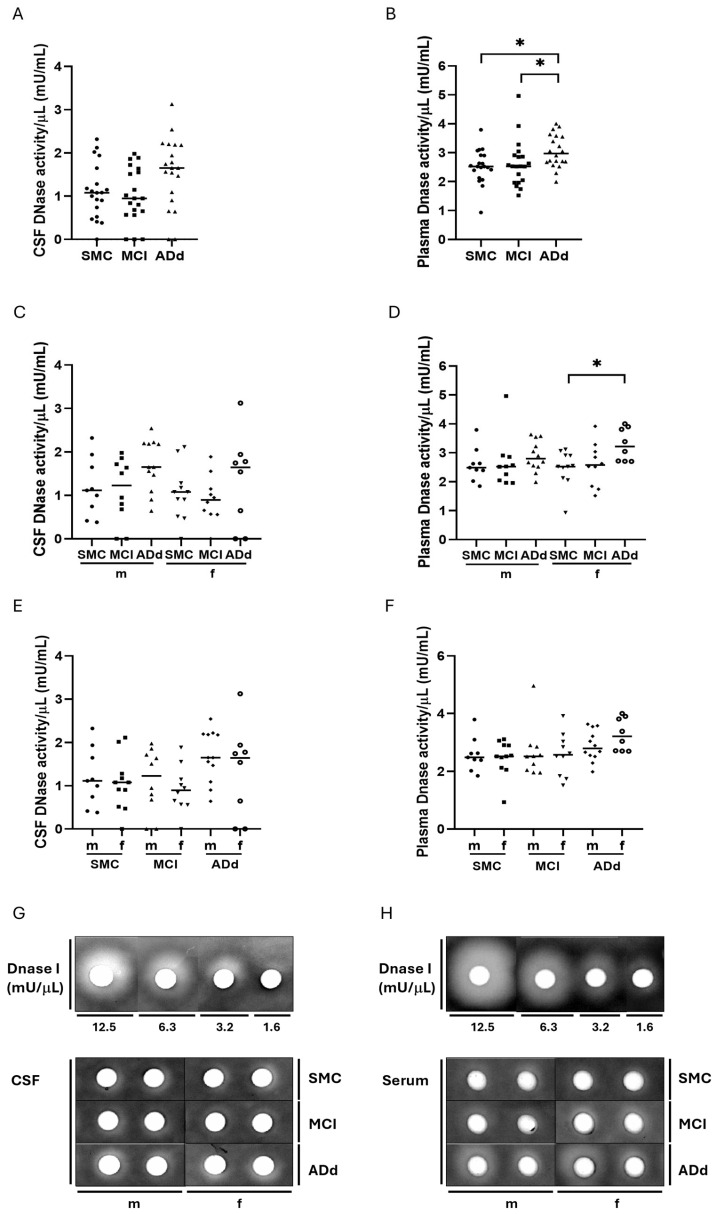
Values of DNase activity in CSF and plasma samples in the three study groups. Data are individual values with medians. Panels (**A**,**B**): DNase activity values in CSF and plasma, respectively, from 20 SMC subjects, 20 MCI subjects, and 20 ADd patients. Panels (**C**,**D**): DNase activity values in CSF and plasma, respectively, according to sex in the three groups. Panels (**E**,**F**): DNase in CSF and plasma, respectively, in subjects dichotomized according to sex in each study group. Panels (**G**,**H**): Representative single radial enzyme diffusion assay. Agarose gels loaded with CSF (**G**) and plasma (**H**) samples, respectively, are shown (See Appendix A). * = *p* < 0.05, Dunn’s post-test. SMC = subjective memory complaints. MCI = mild cognitive impairment. ADd = Alzheimer dementia. CSF = cerebrospinal fluid. m = male group: SMC, *n* = 9; MCI, *n* = 10; ADd, *n* = 12. f = female group: SMC, *n* = 11; MCI, *n* = 10; ADd, *n* = 8.

**Figure 6 ijms-26-07792-f006:**
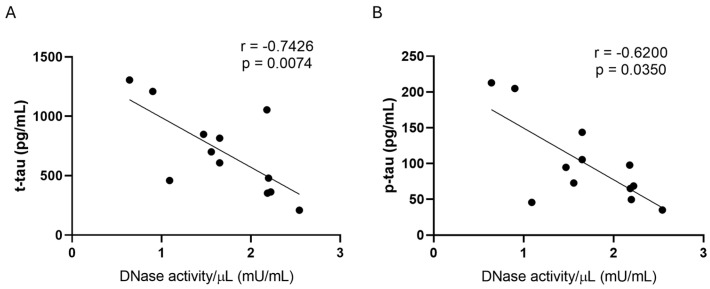
Regression analysis. Spearman’s correlation test between DNase activity and t-tau (panel (**A**)) and p-tau (panel (**B**)) in male ADd patients in CSF samples. *n* = 12.

**Figure 7 ijms-26-07792-f007:**
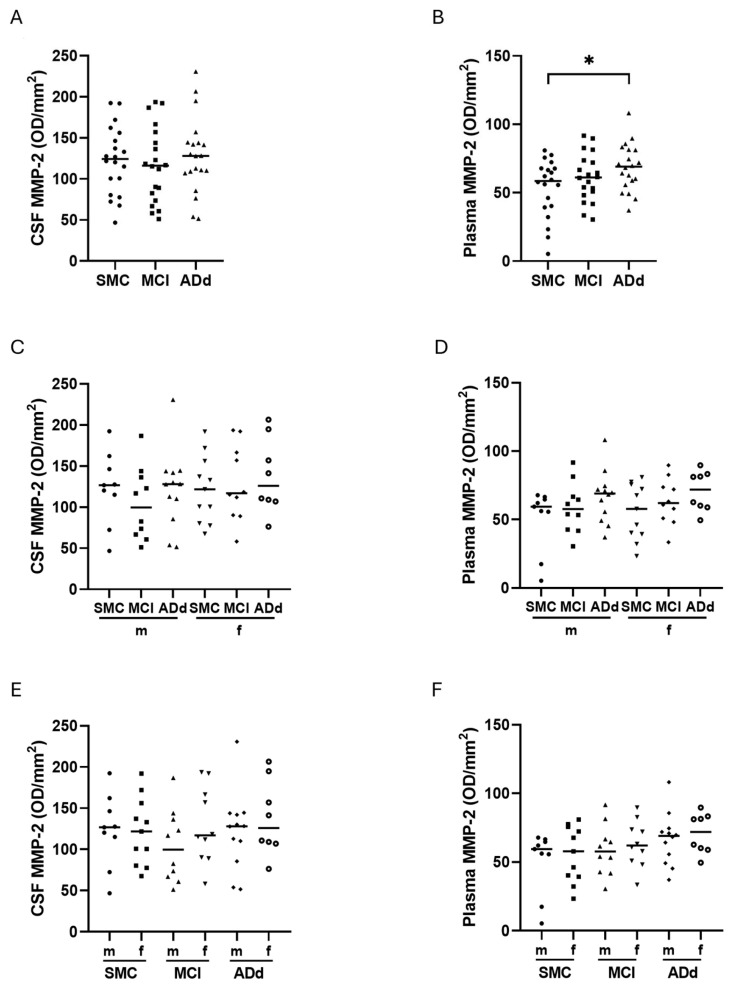
Activity of MMP-2 in CSF and plasma samples in the three subject groups. Data are individual values with medians. Panels (**A**,**B**): in CSF and plasma, respectively, from 20 SMC subjects, 20 MCI subjects, and 20 ADd patients. Panels (**C**,**D**): in CSF and plasma, respectively, according to sex in the three groups. Panels (**E**,**F**): in CSF and plasma, respectively, in subjects dichotomized according to sex in each study group. *: *p* < 0.05, Dunnet’s test. SMC = subjective memory complaints. MCI = mild cognitive impairment. ADd = Alzheimer dementia. CSF = cerebrospinal fluid. m = male group: SMC, *n* = 9; MCI, *n* = 10; ADd, *n* = 12. f = female group: SMC, *n* = 11; MCI, *n* = 10; ADd, *n* = 8.

**Figure 8 ijms-26-07792-f008:**
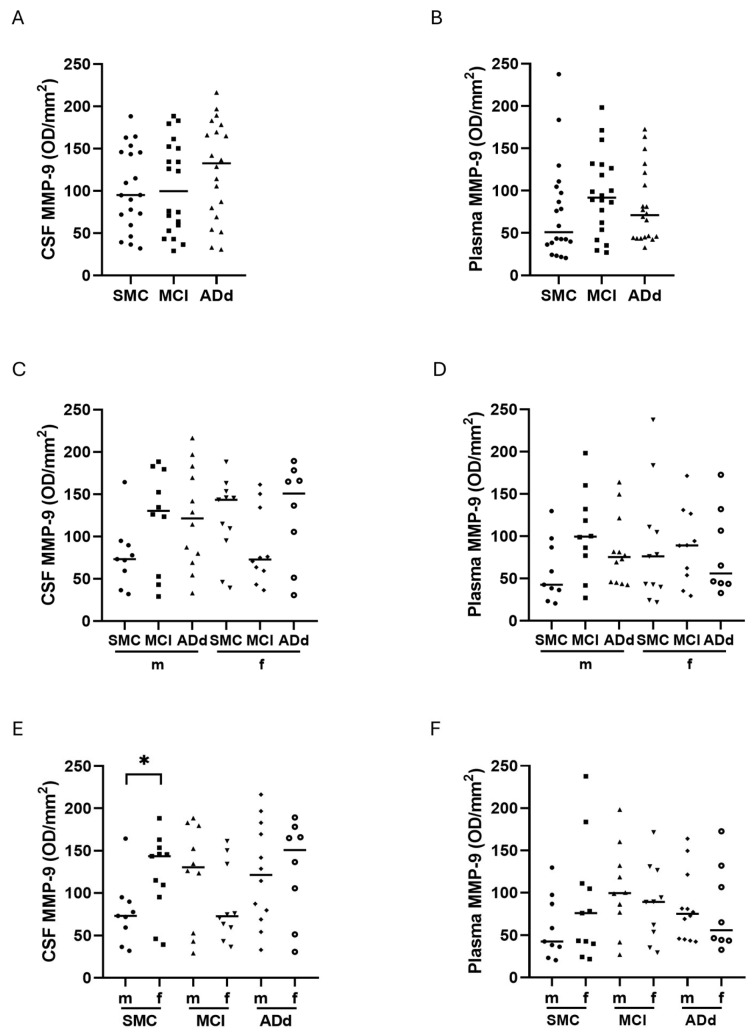
Activity of MMP-9 in CSF and plasma samples in the three subject groups. Data are individual values with medians. Panels (**A**,**B**): in CSF and plasma, respectively, from 20 SMC subjects, 20 MCI subjects, and 20 ADd patients. Panels (**C**,**D**): in CSF and plasma, respectively, according to sex in the three groups. Panels (**E**,**F**): in CSF and plasma, respectively, in subjects dichotomized according to sex in each study group. *: *p* < 0.05, *t*-test. SMC = subjective memory complaints. MCI = mild cognitive impairment. ADd = Alzheimer dementia. CSF = cerebrospinal fluid. m = male group: SMC, *n* = 9; MCI, *n* = 10; ADd, *n* = 12. f = female group: SMC, *n* = 11; MCI, *n* = 10; ADd, *n* = 8.

**Figure 9 ijms-26-07792-f009:**
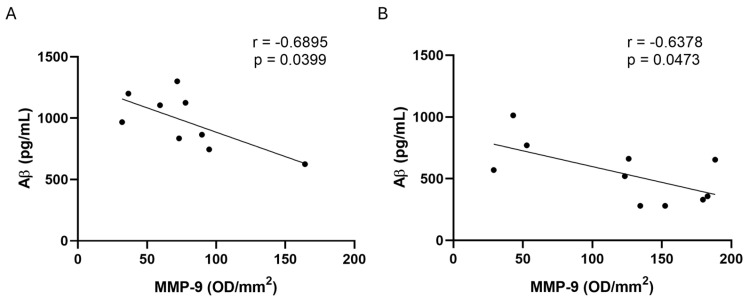
Regression analysis. Spearman’s correlation test between MMP-9 and Ab in male SMC subjects (panel (**A**), *n* = 9) and in male MCI patients (panel (**B**), *n* = 10) in CSF samples.

**Table 1 ijms-26-07792-t001:** Demographic and clinical data of the participants to the study.

	SMC (*n* = 20)	MCI (*n* = 20)	ADd (*n* = 20)	*p*
Sex (Male/Female)	9/11	10/10	12/8	0.6268
Age (years)	60.3 ± 11.2	65.7 ± 9.8	66.7 ± 9.3	0.1130 °
CSF Aβ (pg/mL)	937.7 (829.6–1171.0)	518.0 (375.0–640.7) ^#^	442.0 (364.2–568.7) ^#^	<0.0001 ^§^
Male (pg/mL)	968.7 (790.5–1163.0)	546.0 (317.8–689.3) ^#^	501.9 (346.2–578.0) ^#^	0.0004 ^§^
Female (pg/mL)	910.0 (827.4–1186.0)	511.5 (433.8–560.9) ^#^	389.3 (374.5–520.8) ^#^	<0.0001 ^§^
CSF t-tau (pg/mL)	170.8 (134.2–234.5)	398.5 (184.0–548.5) ^#^	654.2 (357.5–896.7) ^#,@^	<0.0001 ^§^
Male (pg(mL)	175.0 (138.4–217.0)	313.0 (172.3–602.6)	654.2 (387.0–1003.0) ^#^	0.0006 ^§^
Female (pg/mL)	166.7 (116.0–310.0)	404.5 (192.0–575.0)	645.7 (351.4–894.3) ^#^	0.0022 ^§^
CSF p-tau (pg/mL)	28.86 (19.98–36.40)	55.80 (33.03–71,71) ^#^	82.50 (49.25–103.6) ^#^	<0.0001 ^§^
Male (pg/mL)	25.2 (18.8–31.7)	49.3 (35.9–70.2) ^#^	83.8 (53.6–134.1) ^#^	0.0001 ^§^

Age is means ± sd; clinical data are medians and 25–75th interquartile range. *p* = χ^2^ test; ° = One-Way ANOVA; ^§^ = Kruskal–Wallis test. ^#^ = significant versus SMC group. ^@^ = significant versus MCI group. SMC = subjective memory complaints. MCI = mild cognitive impairment. ADd = Alzheimer dementia. Aβ = amyloid Aβ. t-tau = total tau. p-tau = hyperphosphorylated tau.

## Data Availability

Further information related to the current study is available from the corresponding authors upon reasonable request.

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
