# Peer review of "Reliable New Biomarkers of Mitochondrial Oxidative Stress and Neuroinflammation in Cerebrospinal Fluid and Plasma from Alzheimer’s Disease Patients: A Pilot Study"

_ijms, 2025, doi:10.3390/ijms26167792_

Round 1
Reviewer 1 Report
Comments and Suggestions for Authors
The paper "Reliable new biomarkers of mitochondrial oxidative stress and neuroinflammation in cerebrospinal fluid and plasma from Alzheimer’s Disease patients. A pilot study." submitted by Di Lorenzo et al. is significant and of strong interest to the scientific community. As the authors themselves acknowledge, although the experimental design is sound, the sample size is limited, especially considering the high variability typically found in human samples. Nevertheless, it can serve as a starting point for future studies. I also believe that relating marker expression to the patient's sex adds further value. For these reasons, I recommend the publication of the manuscript after addressing the minor issues.
Table 1: which is the meaning of a, b, c and ab close to the values?
In the main text, I cannot find any reference to the supplementary figures.
Author Response
We are deeply thankful to you for the attention and consideration dedicated to our work and we thank the Reviewers for their dedication to review our manuscript. We tried to follow all the Reviewers suggestions in the revised version and to answer to the points raised by them in order to make the paper more complete and reliable in the revised version.
Here follows a point by point reply to the Reviewers’ comments.
REVIEWER 1
Comments and Suggestions for Authors
The paper "Reliable new biomarkers of mitochondrial oxidative stress and neuroinflammation in cerebrospinal fluid and plasma from Alzheimer’s Disease patients. A pilot study." submitted by Di Lorenzo et al. is significant and of strong interest to the scientific community. As the authors themselves acknowledge, although the experimental design is sound, the sample size is limited, especially considering the high variability typically found in human samples. Nevertheless, it can serve as a starting point for future studies. I also believe that relating marker expression to the patient's sex adds further value. For these reasons, I recommend the publication of the manuscript after addressing the minor issues.
Comment 1. - Table 1: which is the meaning of a, b, c and ab close to the values?
Response 1. - We are grateful to the Reviewer for appreciating our work and for his/her very precise comments, which helped us to further improve the quality of our manuscript. Thanks to the Reviewer’s suggestion we have modified in Table 1 the letters symbolizing statistically significant differences between groups with clearer symbols that should make more direct the identification of such differences.
Comment 2. - In the main text, I cannot find any reference to the supplementary figures.
Response 2. - We thank the Reviewer for his/her careful reading of the text and we have inserted, in the revised version, a reference to the supplementary figures at lines 203 and 273 of the legends, respectively, of Figures 1 and 5.
Hoping to have solved the Reviewers’ concerns, we thank you for the consideration dedicated to our work and we send our best regards,
Vito Pesce and Grazia Maria Liuzzi
Reviewer 2 Report
Comments and Suggestions for Authors
Dear Authors,
I have carefully read your manuscript aimed to characterize mitochondrial involvement and neuroinflammation development in the AD progression through the identification of novel mitochondrial and neuroinflammatory biomarkers, which might be added to the routinely assessed ones.
Paper is nicely written, provides novel insights and has clinical significance.
Introduction contains many information on the topic and provides broad background, which is of importance to the readers.
There are certain improvements that should be made.
Materials and methods should be in 2nd section instead of 4th.
Type of study should be added according to STROBE criteria. Also, inclusion/exclusion criteria should be clear and concise (for examples Is ther age limit for each of the groups or not). Variables included in the study are great, especially imaging finding. However, statistical analysis is basic given the small number of patients, being mainly descriptive and correlations.
Results are well presented. However, I suggest also adding imaging findings for each of the patient group – 3 figures containing MR and PET findings.
Discussion is well written, as all paper overall.
Author Response
We are deeply thankful to you for the attention and consideration dedicated to our work and we thank the Reviewers for their dedication to review our manuscript. We tried to follow all the Reviewers suggestions in the revised version and to answer to the points raised by them in order to make the paper more complete and reliable in the revised version.
Here follows a point by point reply to the Reviewers’ comments.
REVIEWER 2
Comments and Suggestions for Authors
Dear Authors,
I have carefully read your manuscript aimed to characterize mitochondrial involvement and neuroinflammation development in the AD progression through the identification of novel mitochondrial and neuroinflammatory biomarkers, which might be added to the routinely assessed ones.
Paper is nicely written, provides novel insights and has clinical significance.
Introduction contains many information on the topic and provides broad background, which is of importance to the readers.
There are certain improvements that should be made.
Comment 1. - Materials and methods should be in 2nd section instead of 4th.
Response 1. - We are grateful to the Reviewer for his/her suggestion, but we followed in the manuscript the order of sections indicated in the very recent Microsoft Word template of International Journal of Molecular Sciences, according to which the Materials and methods section had to be the fourth one and not the second one. It is not in our possibility to change such editorial rules.
Comment 2. - Type of study should be added according to STROBE criteria. Also, inclusion/exclusion criteria should be clear and concise (for examples Is ther age limit for each of the groups or not). Variables included in the study are great, especially imaging finding. However, statistical analysis is basic given the small number of patients, being mainly descriptive and correlations.
Response 2. - We are grateful to the Reviewer for appreciating our work and for his/her valuable and insightful comments, which helped us to definitively improve the quality of our research.
Please, find below our detailed responses to each raised point:
Type of study
We have clarified in the revised manuscript that this is a cross-sectional observational study, in accordance with the STROBE guidelines [von Elm E, Altman DG, Egger M, Pocock SJ, Gøtzsche PC, Vandenbroucke JP; STROBE Initiative. The Strengthening the Reporting of Observational Studies in Epidemiology (STROBE) Statement: guidelines for reporting observational studies. Lancet. 2007 Oct 20; 370(9596):1453–1457)].
This has now been explicitly stated in the Methods section (page 18 line 526) and cited among the references (n. 70): “This is a cross-sectional study according to the STROBE (Strengthening the Reporting of Observational Studies in Epidemiology) reporting guideline [70], including sixty subjects outpatients who consecutively referred to the Center for Neurodegenerative Diseases and the Aging Brain of the University of Study of Bari “Aldo Moro” at Pia Fondazione “Card. Panico” Hospital (Tricase):…“
Inclusion/exclusion criteria
Patients were selected based on their consecutive referral to our Center over a defined time-period, from January 2016 to December 2021, following a temporal continuum and according to diagnostic criteria, as reported in the Methods’ section.
We have included in the main text the patients’ selection period (Methods’ section: page 18, line 530) and in the Results’ section the range of age for each group (page 4, line 171), as follows: “The age range was 39-81 years for SMC, 50-88 years for MCI, 36-79 years for ADd”.
Concerning the exclusion criteria, we also ruled out subjects who depicted poor compliance to the study procedures and with a poor or incomplete clinical history, to avoid any confounding factor. We have also added these comments in the Methods’ section (page 18, line 565): “Subjects who showed poor adherence to study procedures and had a poor or incomplete medical history were also excluded”.
Imaging data and their role in the study
Imaging findings (MRI and PET) were not the focus of this study and were not collected as primary variables for statistical correlation. These data were used solely for diagnostic purposes and were found to be compatible with the diagnosis of Alzheimer's Disease, according to the current international diagnostic criteria for Alzheimer’s Disease (National Institute on Aging-Alzheimer’s Association (NIA-AA) criteria, 2011 and 2018).
Comment 3. - Results are well presented. However, I suggest also adding imaging findings for each of the patient group – 3 figures containing MR and PET findings.
Response 3. - We deeply appreciate the Reviewer’s suggestion. We would like to clarify that this study was not designed to evaluate or compare neuroimaging findings across groups. However, we acknowledge the relevance of neuroimaging and will consider it in future studies specifically focused on imaging analyses.
Hoping to have solved the Reviewers’ concerns, we thank you for the consideration dedicated to our work and we send our best regards,
Vito Pesce and Grazia Maria Liuzzi